# Fetal Growth Associated with Maternal Rheumatoid Arthritis and Juvenile Idiopathic Arthritis

**DOI:** 10.3390/healthcare12232390

**Published:** 2024-11-28

**Authors:** Eugenia Yupei Chock, Bente Glintborg, Zeyan Liew, Lars Henning Pedersen, Mette Østergaard Thunbo

**Affiliations:** 1Section of Rheumatology, Allergy and Immunology, Yale School of Medicine, 300 Cedar Street, New Haven, CT 06520, USA; 2DANBIO and Copenhagen Center for Arthritis Research (COPECARE), Center for Rheumatology and Spine Diseases, Centre of Head and Orthopedics, University Hospital of Copenhagen Rigshospitalet, 2100 Glostrup, Denmark; 3Department of Clinical Medicine, Faculty of Health and Medical Sciences, University of Copenhagen, 1172 Copenhagen, Denmark; 4Department of Environmental Health Sciences, Yale Center for Perinatal, Pediatric, and Environmental Epidemiology, Yale School of Public Health, One Church Street, New Haven, CT 06510, USA; 5Department of Obstetrics and Gynecology, Aarhus University Hospital, Aarhus University, 8000 Aarhus N, Denmark; 6Department of Clinical Medicine, Aarhus University, 8200 Aarhus N, Denmark; 7Department of Clinical Pharmacology, Aarhus University Hospital, 8200 Aarhus N, Denmark

**Keywords:** autoimmune disease, fetal growth, high-risk pregnancies, maternal fetal medicine, rheumatoid arthritis, juvenile idiopathic arthritis

## Abstract

**Introduction:** Patients with rheumatoid arthritis (RA) and juvenile idiopathic arthritis (JIA) are at a twice-higher risk of developing adverse pregnancy outcomes, such as preterm births and infants with a low birth weight. We aimed to evaluate fetal growth among patients with and without rheumatoid arthritis and juvenile idiopathic arthritis (RA and JIA). **Materials and Methods:** We conducted a population-based cohort study in Denmark from 2008–2018, which included 503,491 singleton pregnancies. Among them, 2206 were pregnancies of patients with RA and JIA. We linked several nationwide databases and clinical registries in Denmark to achieve our aim. First, we used the International Classification of Diseases-10 codes to identify pregnant patients with RA and JIA from the National Patient Registry. Next, we obtained fetal biometric measurements gathered from second-trimester fetal ultrasound scans and birthweights through the Fetal Medicine Database. Finally, we computed a fetal growth gradient between the second trimester and birth, using the mean difference in the Z-score distances for each fetal growth indicator. We also calculated the risk of small for gestational age (SGA). All outcomes were compared between pregnant individuals with and without RA and JIA, adjusted for confounders. **Results:** Maternal RA and JIA were not associated with a reduction in the estimated fetal weight (EFW) at 18 to 22 weeks of gestational age [adjusted mean EFW Z-score difference of 0.05 (95% CI 0.01, 0.10)]. We observed reduced mean Z-score differences in the weight gradient from the second trimester to birth among offspring of patients with RA and JIA who used corticosteroids [−0.26 (95% CI −0.11, −0.41)] or sulfasalazine [−0.61 (95% CI −0.45, −0.77)] during pregnancy. Maternal RA and JIA were also associated with SGA [aOR of 1.47 (95% CI 1.16, 1.83)] and the risk estimates were higher among corticosteroid [aOR 3.44 (95% CI 2.14, 5.25)] or sulfasalazine [(aOR 2.28 (95% CI 1.22, 3.88)] users. **Conclusions:** Among pregnant patients with RA and JIA, fetal growth restriction seemed to occur after 18 to 22 weeks of gestational age. The second half of pregnancy may be a vulnerable period for optimal fetal growth in this population.

## 1. Introduction

Rheumatoid arthritis (RA) and juvenile idiopathic arthritis (JIA) are rheumatic diseases that mainly affect the joints and are characterized by systemic inflammation [1,2]. Patients with RA and JIA have poorer pregnancy and birth outcomes compared to those without the conditions [3,4]. The offspring of patients with RA and JIA have almost double the risk of being born preterm; furthermore, infants born to patients with RA are likely to be smaller for gestational age (SGA), although this is observed at a lesser degree among patients with JIA [4,5,6,7]. Offspring impacted by fetal growth restriction may develop chronic medical conditions during their adult lives [8,9]. Placental insufficiency is one of the common causes of reduced fetal growth and a low birth weight [10]. RA has been implicated to directly affect placental growth and function [11,12], although other factors such as medication use and disease activity may also contribute to smaller-sized offspring [11,13]. Prior fetal growth studies among patients with RA focused on birth weights and maternal cytokines influencing fetal growth [14]. Our study aimed to compare fetal growth indicators, specifically estimated fetal weights during the second trimester (18–22 weeks’ gestational age) and birthweights among offspring of individuals with and without RA and JIA. In addition, we also evaluated fetal growth indicators based on the maternal use of conventional synthetic Disease-Modifying Antirheumatic Drugs (csDMARDs).

## 2. Materials and Methods

### 2.1. Data Source

We conducted a population-based cohort study using several administrative and clinical registries in Denmark from the years 2008–2018. Particularly, we linked the Danish Fetal Medicine Database (DFMD), the Danish Medical Birth Registry (DMBR), the Danish National Patient Registry (DNPR), and the Danish National Prescription Registry (NPR) through a unique civil registration number assigned for each Danish resident. The linked registries allow for the identification of medical and sociodemographic records of mother–offspring pairs.

### 2.2. Identification of Study Population and Patient Characteristics

Our study consisted of pregnant individuals identified from DFMD, who had live singleton pregnancies at 22-weeks gestational age and after (Figure 1). All pregnancies of this gestational age cut-off would have received fetal ultrasound scans during the second trimester. As RA can occur during early adulthood and the onset of JIA can be observed around pre-school age, we used ICD-10 codes to identify patients who were diagnosed with RA and JIA 10 years prior, or during their pregnancies from the DNPR (Appendix A). This allowed us to include as many pregnant women with RA and JIA as possible in the study cohort. The positive predictive value of RA diagnosis within the DNPR is 79% [15]. Pregnant individuals who did not have ICD-10 codes of RA and JIA served as the comparator (unexposed) group.

### 2.3. Antirheumatic Therapy Uptake Before and After Second-Trimester Fetal Ultrasound Study

We identified the csDMARDs of interest with Anatomical Therapeutic Chemical (ATC) codes (Appendix A) from the NPR. A patient with RA and JIA was considered exposed to an csDMARD when at least one refill of csDMARD during pregnancy was recorded. A refill is defined as 30 to 60 days’ worth of medications. We described the uptake of csDMARDs before and after second-trimester fetal ultrasound studies (Figure 2). csDMARDs such as methotrexate and leflunomide were excluded as they are contraindicated during pregnancy, and we also did not include non-steroidal anti-inflammatory drugs as these are not recommended during pregnancy at 20-weeks gestational age or later. Biologic Disease-Modifying Antirheumatic Drugs (bDMARDs) were not reported in this study as the NPR only contains information on prescription medications while bDMARDs are provided through hospitals in Denmark, and not available within the NPR.

### 2.4. Ascertainment of Outcomes

The outcomes of interest include fetal growth indicators (fetal weight, birth weight, and head circumference) and infants born SGA.

*Fetal and Birth Weights*: The DFMD contains standard fetal biometric measurements obtained during first- and second-trimester fetal ultrasound examinations of nearly all pregnant individuals in Denmark. Fetal ultrasounds performed during early pregnancy are expected to have >90% completeness covering all pregnant individuals in Denmark since 2008 [16]. As a fetal ultrasound examination during the first trimester is often used for gestational age dating in Denmark, such measure may be biased when studying week-specific growth measurement. Hence, our analysis has a priori chosen to focus on analyzing second-trimester fetal ultrasound data (18–22 weeks of gestational age), which more accurately reflect fetal growth parameters for the correlated gestational age of pregnancy. In Denmark, all fetal ultrasounds are performed by trained and certified sonographers following standard operating procedures, free of charge. We extracted fetal head circumferences, abdominal circumferences, and femur lengths (all in mm) from second trimester fetal ultrasounds of the study population. We then computed the estimated fetal weights (EFW) with the Hadlock formula based on those measurements [17]. We anticipated mostly accurate EFW estimations which are commonly within a 10% error, based on prior studies [18,19]. We subsequently obtained the birth weights of the respective offspring following the second trimester fetal ultrasound.

*Head circumferences (HC)*: we extracted measurements of the offspring HC from the second trimester and birth to assess the direction of head growth in relation to weight.

*Small-for-gestational-age (SGA)*: defined as a birth weight at least 2 standard deviations below the mean for gestational age (corresponding to −2 Z-scores) [20], as an indicator of restricted fetal growth.

For all outcomes, we first compared pregnancies with and without RA and JIA. Next, we stratified the RA and JIA group by different pregnancy-compatible csDMARDs used during pregnancy; bDMARDs were excluded due to the reasons mentioned above.

### 2.5. Covariates

Relevant covariates associated with the outcomes of interest were selected a priori, via the construction of the Direct Acyclic Graph and literature review. All analyses for outcomes were adjusted for maternal age at conception, pre-pregnancy BMI, smoking status, income status, race, parity, birth year, pre-pregnancy hypertension, pre-pregnancy diabetes, and co-medication use. Sources of several covariates were derived from the registries noted in Figure 1. As Denmark has an extensive social security and high-income equality system, we consider the income status obtained from the Employment Classification Module (ECM) as an indicator of the socioeconomic status. The ECM contains the main employment activities of Danish residents within each calendar year. Co-medications were defined as dispensed medications not listed in Appendix A; hospital-administered medications were not available within the NPR. All other variables such as smoking status were derived from the DFMD.

### 2.6. Statistical Analysis

We performed descriptive statistics to report the general characteristics of the study population. We then converted EFW and HC to Z-scores using means and standard deviations from international and published standardized growth measures stratified by the gestational week [17,21]. To understand the direction of fetal growth, we calculated the gradient of weight and HC differences between the second trimester and birth of the offspring. The formula to calculate the gradient in the weight and HC differences between the second trimester and at birth was as follows:mean difference [(weight or HC Z-score at birth) − (EFW or HC Z-score during second trimester)]

Negative values indicate reduced growth between the second trimester and birth, compared to offspring born to individuals without RA and JIA. We performed multivariable linear regression analyses to estimate the mean differences in the fetal weight, birth weight, weight gradient, head circumference, and head circumference gradient (all in Z-scores), according to maternal RA and JIA diagnoses and their csDMARD use during pregnancy. We conducted multiple logistic regression to estimate the Odds Ratio (OR) and a 95% confident interval for SGA among the offspring associated with maternal RA and JIA and the respective maternal ART exposures. All analyses were adjusted for confounding variables, namely maternal age, maternal body mass index, smoking status, income status, race, parity, birth year, pre-pregnancy hypertension, pre-pregnancy diabetes, and co-medication use during pregnancy. 

### 2.7. Mediation Analysis

Finally, we performed causal mediation analysis to evaluate the degree to which pre-eclampsia mediates the association between pregnancy exposed to RA and JIA and the risk of SGA among the offspring, adjusted for the covariates noted above. We performed this with the *CMAverse* package in RStudio. We estimated the OR of the total effect and controlled direct effect between exposure and the outcome in the mediation analysis model.

This cohort study was approved by the Danish Data Protection Agency (Aarhus University journal number 2016-051-000001, serial number 2227, 2 March 2021) and the Institutional Review Boards of Yale University (IRB ID: 2000034411). By Danish law, no informed consent was required because this was a register-based study using anonymized data. This study followed the Strengthening the Reporting of Observational Studies in Epidemiology (STROBE) reporting guideline [22]. All analyses were performed with RStudio version 2022.12.0. (RFoundation, Vienna, Austria or Posit, Boston, MA, USA).

## 3. Results

The study population included 503,491 singleton pregnancies (the study selection process is depicted in Figure 1). In total, there were 2206 pregnancies of patients with RA (N = 1897, 86.0%) and JIA (N = 588, 26.7%). Compared to pregnant individuals without RA and JIA, patients with these conditions were more likely to be older than 35 years of age at the time of pregnancy (21.7% vs. 16.8%), had a BMI between 25 and 29 (24.2% vs. 21.8%), and experienced more pre-term births during 32 and <37 weeks of gestational age (5.8% vs. 4.1%). Patients with RA and JIA were likelier to have co-morbidities before and during pregnancy; this includes pre-gestational hypertension (1.9% vs. 0.9%), pre-gestational diabetes (1.2% vs. 0.7%), gestational hypertension (3.1% vs. 2.2%), and pre-eclampsia and eclampsia (4.9% vs. 3.9%). Moreover, 25.5% of patients with RA and JIA underwent a cesarean section, compared to those without (20.0%). Patients with RA and JIA were also of a lower socioeconomic status (28.9% on a transfer payment and in early retirement vs. 22.7%). The proportion of women with gestational diabetes were the same for both groups (3.6%). Both groups were also similar in terms of race (>90% Caucasian) and smoking status (non-smoker 89.8% vs. 89.4%) (Table 1).

When assessing the EFW Z-scores obtained from fetal US biometric measurements during the second trimester (Appendix A), the overall adjusted mean EFW Z-score difference among the offspring of patients with RA and JIA was 0.05 compared to those unexposed (95% CI 0.01, 0.10) (Appendix A). When stratified by csDMARD, patients with RA and JIA who used sulfasalazine (SSZ) had an adjusted mean EFW Z-score difference of 0.38 (95% CI 0.24, 0.50). For the birth weight, the overall adjusted mean Z-score difference among the offspring of patients with RA and JIA was −0.08 (95% CI −0.13, −0.04) (Appendix A). The adjusted mean Z-score difference for women with RA and JIA who used corticosteroids (CCS) during pregnancy, compared to those without RA and JIA, was −0.31 (95% CI −0.43, −0.18); −0.23 (95% CI −0.37, −0.10) among SSZ users; and −0.27 (95% CI −0.59, 0.04) among HCQ users (Appendix A).

When assessing fetal growth by analyzing the gradient in the mean differences of weight and head circumferences between the second trimester and birth, the adjusted Z-score mean difference between EFW and birth weight was −0.14 (95% CI −0.08, −0.19;) for the offspring of patients with RA and JIA compared with no RA and JIA (Figure 3). Patients with RA and JIA who used CCS during pregnancy had an adjusted EFW and birth weight difference of −0.26 (95% CI −0.11, −0.41), −0.25 (95% CI −0.64, 0.13) for hydroxychloroquine (HCQ) users, and −0.61 (95% CI −0.45, −0.77) for SSZ use during pregnancy, compared with the offspring of patients without RA and JIA (Figure 3). The overall adjusted Z-score difference of head circumferences between the second trimester and birth was −0.03 (95% CI −0.09, 0.02; *p* = 0.2), comparing RA and JIA to no RA and JIA. The adjusted Z-score differences in head circumferences between the second trimester and birth for patients with RA and JIA and who took CCS during pregnancy was −0.19 (95% CI −0.02, −0.36; *p* = 0.026), −0.33 (95% CI −0.74, 0.09; *p* = 0.12) for HCQ, and −0.20 (95% CI −0.02, −0.37; *p* = 0.03) for SSZ users during pregnancy (Figure 4).

The offspring born to patients with RA and JIA had a higher risk of SGA compared with no RA and JIA (aOR 1.47, 95% CI 1.16, 1.83) (Table 2). The adjusted OR for SGA stratified by ART use among patients with RA and JIA were as follows: CCS [aOR 3.44 (95% CI 2.14,5.25)], HCQ [aOR 3.12 (95% CI 0.74, 8.80)], and SSZ [aOR 2.28 (95% CI 1.22, 3.88]. In mediation analysis, pre-eclampsia was estimated to mediate about 4.3% of the total effect between prenatal RA and JIA exposure and offspring risk for SGA (Appendix A).

## 4. Discussion

To our knowledge, this is the first population-based cohort study that has investigated fetal growth with fetal ultrasound biometric data among the offspring of patients with RA and JIA. Overall, we found a reduction in fetal growth after 18 to 22 weeks of gestational age among pregnant patients with RA and JIA. The EFW among the offspring of patients with RA and JIA during the second trimester were slightly higher compared to those without RA and JIA, notably the EFW were highest among SSZ users during pregnancy. Interestingly, the finding did not persist up to birth as birth weights among the offspring of patients with RA and JIA were lower than the non-RA and -JIA group. The greatest reduction in the fetal growth gradient between the second trimester and birth were among RA and JIA patients who used CCS and SSZ during pregnancy. We also found a simultaneous reduction in head growth between the second trimester and birth.

Fetuses of patients with RA and JIA who used CCS or SSZ during pregnancy had higher EFW during the second trimester compared to fetuses of patients without RA and JIA. As patients with RA and JIA are at higher risk of early pregnancy losses and subfertility [23,24], it is plausible that the finding of higher EFW during the second trimester among women with RA and JIA was influenced by survivor bias. This is a situation in which offspring exposed to maternal RA and JIA were not likely to survive until birth and many pregnancy losses may have already occurred prior to the second-trimester fetal ultrasound study [25]. Despite the overall higher EFW among patients with RA and JIA in our study, the pattern did not persist up to birth as infants born to these groups also experienced reduced growth after the second trimester, compared to infants born to patients without RA and JIA. The findings on CCS use during pregnancy among patients with RA and JIA were concurrent with the studies carried out on the general population, in that antenatal CCS use is highly associated with reduced fetal growth and a low birth weight [26,27]. The gradient of fetal growth from the second trimester to birth was most reduced among RA and JIA patients who used SSZ during pregnancy. We observed a modest drop in the refills of both medications after second-trimester fetal ultrasounds were performed, indicating that the reduction in fetal growth may be exposure-dependent and the cessation of use is unlikely to reverse the phenomenon. Of note, SSZ is generally recommended for the treatment of RA during pregnancy; it has shown promise in the treatment of pre-eclampsia and pre-term labor in both in vivo and in vitro studies [11,28]. Therefore, the finding of a reduced birth weight among SSZ users remains unclear. A plausible explanation would be that high RA and JIA clinical disease activity, which necessitates the use of CCS and SSZ, is likely responsible for the negative effect on fetal growth. This is supported by prior studies that have reported an association between adverse pregnancy outcomes and high clinical disease activity in patients with inflammatory arthritis [7,13,29,30].

In addition to the findings above, several results were consistent with prior studies on pregnancy outcomes among patients with RA and JIA: pregnant patients with RA tend to be older at the time of pregnancy, at a higher risk for hypertensive disorders during pregnancy, tend to undergo cesarean section deliveries, and their offspring are likelier to have restricted fetal growth [13,31,32]. Patients with RA and JIA in our study had higher rates of cesarean sections and pre-eclampsia [6,24,32,33]; however, the rates of stillborns were the same as individuals without RA and JIA [6]. From our mediation analysis, we found that pre-eclampsia accounted for only about 4% of fetal growth restriction, indicating that other factors (i.e., placental changes unrelated to pre-eclampsia) are likely responsible.

We postulate that the period of pregnancy after 18–22 weeks of gestational age may be most vulnerable for optimal growth and development among the offspring of pregnant RA AND JIA patients. Although pre-eclampsia may have a minor mediating role in SGA within this population, additional antenatal factors such as antenatal bDMARD use and clinical disease activity among patients with RA and JIA during this critical period would be important to examine. RA and JIA patients who experience high clinical disease activity or flares are usually treated with CCS [34,35]. As our study findings are concurrent with prior investigations on the adverse impact of antenatal CCS use against fetal growth, treating physicians should make every effort to control clinical disease activity during early pregnancy and minimize antenatal CCS use. It is also possible that among pregnant patients with RA and JIA, cytokine interactions are occurring at the maternofetal interface, resulting in placental dysfunction and leading to decreased fetal weight during later the part of pregnancy [11,36]. We did not find a meaningful association between antenatal HCQ use with the outcomes of interest, but the number of RA and JIA patients on HCQ during pregnancy was small (N = 43).

Closer monitoring and follow-ups between the second trimester and birth may be considered in the routine antenatal care of patients with RA and JIA. This may include third-trimester fetal ultrasound studies and the monitoring of RA and JIA disease activity. As this is an observational study, further research strategies using randomized controlled trials or target trial emulation would illuminate the causal effect of medication use vs. disease activity on fetal growth in pregnancies of patients with inflammatory arthritis. In addition, an interdisciplinary approach with clear communication between the treating rheumatologist and obstetrician is encouraged. As suggested by prior studies, our findings further solidify the recommendation that treating rheumatologists should confer with their RA and JIA patients with regards to pregnancy planning and the management of disease activity during early pregnancy [34,37].

Our study provided insight into fetal growth at two pregnancy time points, namely the second trimester and at birth. Further fetal biometric measurements via ultrasound during the first and third trimester of pregnancy would provide more accurate fetal growth trajectories among patients with RA and JIA. This is critical to identify the time point at which fetal growth is most compromised during pregnancy, and further aid in investigating maternal or placental factors that occurred during that specific period. We also need granular information such as RA and JIA disease activities and biologic ART use around when fetal ultrasounds are performed.

Our findings have also opened the door for further mechanistic studies on the role of placental growth and development among patients with RA and JIA. As the majority of fetal weight gain occurs during the third trimester of pregnancy, both internal and external insults among patients with RA and JIA may result in placental insufficiency at the later stages of pregnancy, hence compromising fetal development during that period [38,39].

### Strengths and Limitations

This is a large, population-based cohort study that included all singleton pregnancies ≥ 22-weeks gestational age born in Denmark during the 11-year period from 2008 to 2018. A prior study by Rom et. al. reported growth measurements at birth among offspring born to parents with RA [14]. Our study distinguishes itself by providing more details on fetal growth measurements obtained via fetal ultrasound studies during the second trimester. We also examined csDMARD use before and after second-trimester fetal ultrasound studies and investigated if fetal growth differed by csDMARD use. We used medication refill records instead of prescription records. Furthermore, we performed a mediation analysis to understand the excess risk of SGA among RA and JIA patients that pre-eclampsia may contribute to.

Our study has several limitations; this includes a lack of additional fetal ultrasound data from the first and third trimesters of pregnancy. As second-trimester fetal ultrasounds are routinely performed in Denmark as part of antenatal care, these data are readily available and are >80% complete [16]. Comparatively, we expect missing and incomplete data in the first- and third-trimester fetal ultrasounds, creating complexity in modeling trajectories. In addition, first-trimester fetal ultrasound exams are mostly indicated for gestational age determination and not fetal sizes. However, we recognize the importance of understanding fetal growth trajectories among the offspring of this population beyond the second trimester and we encourage future research endeavors to address this. Secondly, there is an incomplete ascertainment of csDMARD used during pregnancy among patients with RA and JIA, e.g., the NPR did not contain information on bDMARDs prescribed to patients. Based on our registry data, we also lacked information on clinical disease activity and were unable to correlate the use of CCS with clinical disease activity measurements. Therefore, we are uncertain if antenatal CCS use was indicated for RA and JIA flares or other reasons. The accuracy of EFW as a fetal growth indicator is influenced by various factors, which includes potentially incorrect gestational ages, fetuses that are undergoing severely restricted growth or macrosomia, and congenital anomalies among others [40,41]. This can be remedied in future studies by including more fetal ultrasound measurements during the third trimester.

For the purpose of this study, the mean differences in the Z-score for the fetal weight and head circumferences between the second trimester and birth was an optimal measurement to understand the growth direction between the two time points. As we include additional fetal ultrasound measurements in future studies, a hierarchical Bayesian model may be best suited to investigate the effect of maternal exposures on fetal growth, while considering repeated measurements of the same individual fetus.

## 5. Conclusions

We observed a reduction in fetal growth in ensuing second-trimester ultrasound during pregnancy, resulting in smaller infants among patients with RA and JIA. This suggests that the second half of pregnancy may be a vulnerable period for optimal fetal growth among pregnant patients with RA and JIA, or that placental changes affect fetal growth in the later part of pregnancy. Therefore, future research should focus on building fetal growth trajectories by including additional fetal ultrasound measurements in later pregnancy among patients with RA and JIA.

## Figures and Tables

**Figure 1 healthcare-12-02390-f001:**
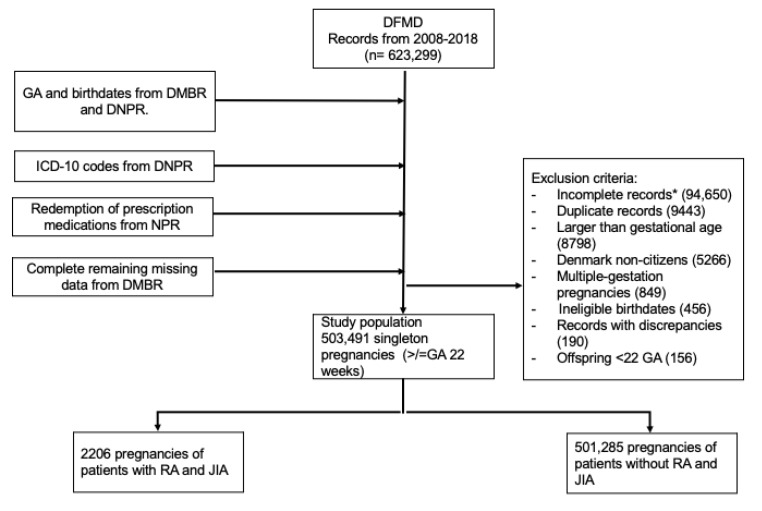
Flowchart of patient selection from linked clinical registries. Legends: DFMD: Danish Fetal Medicine Database; DCRS: Danish Civil Registration System; DMBR: Danish Medical Birth Registry; DNPR: Danish National Patient Registry; NPR: National Prescription Registry; RA: rheumatoid arthritis; JIA: juvenile idiopathic arthritis. * Incomplete records: pregnancy and birth records without gestational ages, lack of second-trimester fetal ultrasound information, lack of birthweights.

**Figure 2 healthcare-12-02390-f002:**
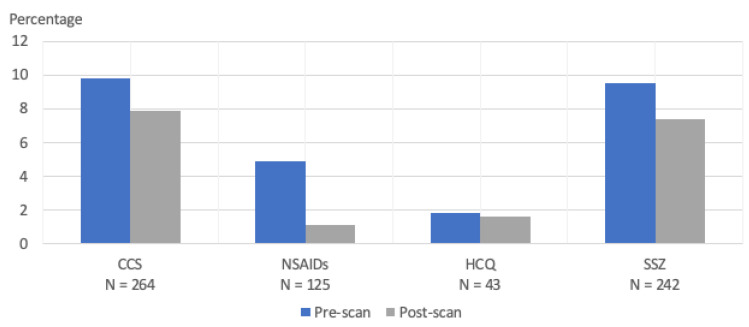
Proportion of antirheumatic therapy uptake at any time before and after second-trimester fetal ultrasound during pregnancy. CCS: corticosteroids; NSAIDs: Non-steroidal anti-inflammatory drugs; HCQ: hydroxychloroquine; SSZ: sulfasalazine.

**Figure 3 healthcare-12-02390-f003:**
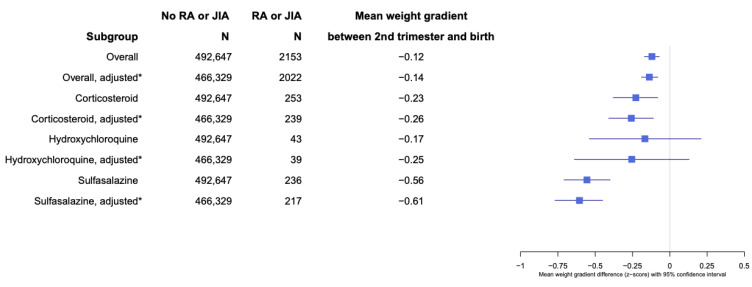
Weight gradient of offspring born to patients with RA and JIA between second trimester and at birth. * Adjusted for maternal age, maternal body mass index, smoking status, income status, race, parity, birth year, pre-pregnancy hypertension, pre-pregnancy diabetes, and co-medication use during pregnancy.

**Figure 4 healthcare-12-02390-f004:**
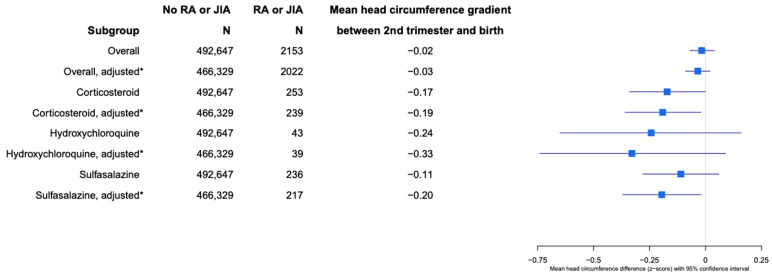
Head circumference gradient of offspring born to patients with RA and JIA between second trimester and at birth. * Adjusted for maternal age, maternal body mass index, smoking status, income status, race, parity, birth year, pre-pregnancy hypertension, pre-pregnancy diabetes, and co-medication use during pregnancy.

**Table 1 healthcare-12-02390-t001:** Baseline characteristics of study population.

Characteristics *	Overall (N = 503,491)	No RA or JIA (N = 501,285)	RA and JIA (N = 2206)
	N (%)	N (%)	N (%)
**Age (years)**			
<25	76,972 (15.3)	76,608 (15.3)	364 (16.5)
25–29	175,596 (34.9)	174,943 (34.9)	653 (29.6)
30–34	166,247 (33.0)	165,536 (33.0)	711 (32.2)
≥35	84,676 (16.8)	84,198 (16.8)	478 (21.7)
**Race**			
Afro-Caribbean	5785 (1.2)	5770 (1.2)	15 (0.7)
Asian	17,230 (3.5)	17,189 (3.5)	41 (1.9)
Caucasian	457,274 (93.6)	455,221 (93.6)	2053 (96.1)
Other ^±^	8436 (1.7)	8409 (1.7)	27 (1.3)
**Pre-gestational Body Mass Index (kg/m^2^)**			
<18.5	31,927 (6.5)	31,798 (6.5)	129 (6.1)
18.5–24	287,290 (58.8)	286,083 (58.8)	1207 (56.7)
25–29	106,537 (21.8)	105,022 (21.8)	515 (24.2)
≥30	62,884 (12.9)	62,607 (12.9)	277 (13.0)
**Smoking status during pregnancy**			
Former	7642 (1.5)	7606 (1.5)	36 (1.7)
Non-smoker	442,531 (89.4)	440,591 (89.4)	1940 (89.8)
Smoker	44,832 (9.1)	44,648 (9.0)	184 (8.5)
**Income status**			
In school	33,471 (6.7)	33,317 (6.6)	154 (7.0)
Employed	351,448 (69.8)	350,043 (69.8)	1405 (63.7)
Transfer payment ^¶^	111,390 (22.1)	110,811 (22.1)	579 (26.2)
Early retirement	3251 (0.6)	3191 (0.6)	60 (2.7)
Unemployed	3699 (0.7)	3691 (0.7)	8 (0.4)
**Parity**			
Nulliparous	180,161 (35.8)	179,317 (35.8)	844 (38.3)
Multiparous	323,324 (64.2)	321,962 (64.2)	1362 (61.7)
**Co-morbidities**			
Pregestational hypertension	4472 (0.9)	4431 (0.9)	41 (1.9)
Gestational hypertension	10,856 (2.2)	10,778 (2.2)	68 (3.1)
Pre-eclampsia/eclampsia	19,814 (3.9)	19,707 (3.9)	107 (4.9)
Pregestational diabetes	3611 (0.7)	3585 (0.7)	26 (1.2)
Gestational diabetes	18,031 (3.6)	17,951 (3.6)	80 (3.6)
**Birth-related characteristics**			
**Pregnancy Outcome**			
Stillborn	1049 (0.2%)	1045 (0.2%)	4 (0.2%)
**Gestational Age at Birth (weeks)**			
<32	3606 (0.7%)	3587 (0.7%)	19 (0.8%)
≥32 to <37	20,488 (4.1%)	20,361 (4.1%)	127 (5.8%)
≥37 to <42	464,218 (92.2%)	462,218 (92.2%)	2000 (90.7%)
≥42	15,179 (3.0%)	15,119 (3.0%)	60 (2.7%)
Mode of Delivery			
Vaginal delivery	402,700 (80.0%)	401,056 (80.0%)	1644 (74.5%)
Cesarean section	100,791 (20.0%)	100,229 (20.0%)	562 (25.5%)

* Variables contain missing values. ^¶^ Transfer payment: Temporary income for individuals administered by employer or government due to a health condition. ± Identify as multiracial.

**Table 2 healthcare-12-02390-t002:** Risk estimates of small-for-gestational-age infants among patients with RA and JIA, stratified by ART use during pregnancy.

Patient Subgroups	Proportion with SGA (%)	Crude Model			Adjusted Model *		
		Odds Ratio (OR)	95% CI	*p*-Value	OR	95% CI	*p*-Value
No RA and JIA	13,025/501,285 (2.6)	1.00	Reference		1.00	Reference	
RA and JIA	87/2206 (3.9)	1.54	1.23, 1.90	**<0.001**	1.47	1.16, 1.83	**<0.001**
- CCS	24/264 (9.1)	3.75	2.40, 5.58	**<0.001**	3.44	2.14, 5.25	**<0.001**
- HCQ	3/43 (7.0)	2.81	0.68, 7.74	0.084	3.12	0.74, 8.80	0.062
- SSZ	14/242 (5.8)	2.30	1.28, 3.80	**0.002**	2.28	1.22, 3.88	**0.005**

* Adjusted for maternal age, maternal body mass index, smoking status, income status, race, parity, birth year, pre-pregnancy hypertension, pre-pregnancy diabetes, and co-medication use during pregnancy. CCS: Corticosteroids; HCQ: hydroxychloroquine; SSZ: sulfasalazine. Bold values denote statistical significance at the *p* < 0.05 level.

## Data Availability

The datasets presented in this article are not readily available because of data protection laws enforced by Danish authorities. Requests to access the datasets should be directed to Danish health authorities (https://www.rkkp.dk).

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
