# Peer review of "Fetal Growth Associated with Maternal Rheumatoid Arthritis and Juvenile Idiopathic Arthritis"

_healthcare, 2024, doi:10.3390/healthcare12232390_

Round 1

Reviewer 1 Report

Comments and Suggestions for Authors

The manuscript"Fetal Growth Associated with Maternal Rheumatoid Arthritis and Juvenile Idiopathic Arthritis," addresses an important association of autoimmune disorders and maternal-fetal health, focusing on how maternal rheumatoid arthritis (RA) and juvenile idiopathic arthritis (JIA) may impact fetal development. However, the manuscript has few areas of scope for improvement to address clarity and in-depth analysis. I have summarized the critical review of the manuscript below. 

·       The manuscript title effectively reflects the topic. Using "RA/JIA" throughout the text may give the impression that rheumatoid arthritis (RA) and juvenile idiopathic arthritis (JIA) are one entity. For clarity, it would be preferable to refer to them distinctly as “RA and JIA,” especially since their clinical implications can vary significantly.

·       The study leverages a robust dataset, utilizing multiple administrative and clinical registries in Denmark from 2008 to 2018. However, details about the underlying population demographics of Denmark would help readers understand the sample’s relevance to broader settings. This contextual information would be valuable for interpreting the applicability of these findings outside of Denmark.

·       The selection criteria involving patients diagnosed with RA or JIA up to 10 years before or during pregnancy raise questions. Clarifying the rationale for the 10-year window (e.g., due to disease progression or potential long-term medication effects) would enhance the understanding of the patient selection process.

·       The study’s focus on fetal biometric data, specifically head circumference, abdominal circumference, and femur length in the second trimester (18-22 weeks), is insightful but seems limited. Expanding the measurements to include the first trimester could provide a more comprehensive picture of growth patterns across gestation.

·       In defining restricted fetal growth, the manuscript uses the term “Small-for-Gestational Age (SGA),” identifying birth weight below 2 standard deviations for gestational age as a marker. However, the term SGA typically encompasses infants who are constitutionally small but healthy. Consider using “Intrauterine Growth Restriction (IUGR)” for clarity, as it more accurately reflects restricted growth due to pathological conditions, like those potentially induced by RA or JIA.

·       Reporting the proportion of RA and JIA cases in percentages would add clarity. Specifically, how many pregnant individuals in the cohort had RA versus JIA would provide a more comprehensive context.

  • The manuscript reports a higher incidence of preterm births (32-<37 weeks) in RA/JIA pregnancies (5.8% vs. 4.1%), yet it is unclear if this difference is statistically significant. Including a significance test here would bolster the strength of the findings and help in evaluating the clinical relevance.
  • The authors state that this is the first cohort study examining fetal growth in RA/JIA pregnancies using ultrasound data. However, previous studies, such as the one by Rom et al. (2014, https://doi.org/10.1002/art.38874 ) with a larger cohort size of 13,556 children, should be acknowledged, as it explored birth anthropometry. Although that study didn’t assess detailed ultrasound measurements, comparisons could be made to highlight the novelty of this study’s use of second-trimester abdominal circumference, an important indicator of fetal growth restriction (FGR). Additionally, limitations include the focus on weight and head circumference at birth without further analysis of other measurements like abdominal circumference, which could strengthen conclusions about FGR patterns in RA and JIA pregnancies. 

Author Response

Comment 1: The manuscript title effectively reflects the topic. Using "RA/JIA" throughout the text may give the impression that rheumatoid arthritis (RA) and juvenile idiopathic arthritis (JIA) are one entity. For clarity, it would be preferable to refer to them distinctly as “RA and JIA,” especially since their clinical implications can vary significantly.

Response 1: Thank you for pointing this out, we agree with this comment. Therefore, we have now changed RA/JIA to RA and JIA throughout the manuscript.

Comment 2: The study leverages a robust dataset, utilizing multiple administrative and clinical registries in Denmark from 2008 to 2018. However, details about the underlying population demographics of Denmark would help readers understand the sample’s relevance to broader settings. This contextual information would be valuable for interpreting the applicability of these findings outside of Denmark.

Response 2: Thank you for the comment, the demographic information of the study cohort is available under “Section 3: Results”, pages 5 and 6, lines 223-246, and in Table 1. As the Danish clinical registries are population-based, the demographic information is reflective of the country’s residents, at-large. The findings from the study have biological relevance that can be applicable in other populations.

Comment 3: The selection criteria involving patients diagnosed with RA or JIA up to 10 years before or during pregnancy raise questions. Clarifying the rationale for the 10-year window (e.g., due to disease progression or potential long-term medication effects) would enhance the understanding of the patient selection process.

Response 3: Thank you for the comment. RA can occur during early adulthood and the onset of JIA can be observed around pre-school age.1,2 Our selection strategy aimed to capture as many RA and JIA patients within the study period. We now clarify this in page 2, lines 89-93.

Comments 4: The study’s focus on fetal biometric data, specifically head circumference, abdominal circumference, and femur length in the second trimester (18-22 weeks), is insightful but seems limited. Expanding the measurements to include the first trimester could provide a more comprehensive picture of growth patterns across gestation.

Response 4: Thank you for this comment. While having fetal biometric data on the first trimester can be useful, there is complexity on including first trimester data in our study. As first trimester fetal ultrasound is used for gestational age dating in Denmark, such measure may be biased when studying week-specific growth measurement (e.g., smaller babies could lead to an adjustment or correction of the gestational age dating based on the self-reported last menstrual period). Hence, our analysis has a priori chosen to focus on analyzing second trimester fetal ultrasound data (18-22 weeks), which more accurately reflects fetal growth parameters for the correlated gestational age of pregnancy. We have now clarified this point in page 4, lines 145-151. At the present stage, we have limited resources, and our protocol precluded us from extracting the first trimester fetal ultrasound data for analysis in this considerably large study population. Despite this, we contend that our study is the first to detail second trimester fetal ultrasound biometric measurements among offspring of women with RA and JIA, using a nationwide study approach. Our findings have important scientific value and is the first study that details fetal ultrasound biometric measurements among offspring of women with RA and JIA. Overall, we do agree with the reviewer that including and exploring fetal biometric measurements from other trimesters is an important next step. We have stated this in our manuscript (page 10, lines 393 to 395), now we further highlight this point as a limitation on page 11, lines 438-446 and in the Conclusion of our paper (page 12, lines 478-480).

Comment 5: In defining restricted fetal growth, the manuscript uses the term “Small-for-Gestational Age (SGA),” identifying birth weight below 2 standard deviations for gestational age as a marker. However, the term SGA typically encompasses infants who are constitutionally small but healthy. Consider using “Intrauterine Growth Restriction (IUGR)” for clarity, as it more accurately reflects restricted growth due to pathological conditions, like those potentially induced by RA or JIA.

Response 5: Thank you for the comment. We agree with the reviewer that the term SGA does not distinct between Ð…GA infants who are constitutionally small and otherwise healthy from those who are small due to growth restriction. Since we did not aim to investigate this distinction in our study, we decided to use the term SGA to accurately reflect the outcome measured. Our study’s current definition of SGA, as it stands, would overestimate the true number of offspring who experienced fetal growth restriction or IUGR. In addition, IUGR is typically defined by two or more fetal ultrasound biometric measurements. To keep the accuracy of the measured outcome, we did not make any changes on the manuscript.

Comment 6: Reporting the proportion of RA and JIA cases in percentages would add clarity. Specifically, how many pregnant individuals in the cohort had RA versus JIA would provide a more comprehensive context.

Response 6: Thank you for the comment. We provided the proportion of RA and JIA cases in page 5, lines 224-225, “In total, there were 2,206 pregnancies of patients with RA (N=1,897, 86.0%) and JIA (N=588, 26.7%).”

Comment 7: The manuscript reports a higher incidence of preterm births (32-<37 weeks) in RA/JIA pregnancies (5.8% vs. 4.1%), yet it is unclear if this difference is statistically significant. Including a significance test here would bolster the strength of the findings and help in evaluating the clinical relevance.

Response 7: Thank you for the feedback. The descriptive table is intended to summarize group characteristics without implying statistical testing, as our study’s primary goal is not to assess specific hypotheses about these baseline differences. Including a p-value here might suggest undue emphasis on additional hypothesis testing and statistical significance, rather than focusing on descriptive nature of the differences. There is increasing call to avoid including inference statistics when describing the data.3 Following these more recent reporting recommendations, we have chosen to report raw percentages without including statistical testing that is not part of the main study hypothesis.

Comment 8: The authors state that this is the first cohort study examining fetal growth in RA/JIA pregnancies using ultrasound data. However, previous studies, such as the one by Rom et al. (2014, https://doi.org/10.1002/art.38874) with a larger cohort size of 13,556 children, should be acknowledged, as it explored birth anthropometry. Although that study didn’t assess detailed ultrasound measurements, comparisons could be made to highlight the novelty of this study’s use of second-trimester abdominal circumference, an important indicator of fetal growth restriction (FGR).

Response 8: Thank you for the comment. We mentioned the study by Rom et. al. in page 11, lines 430- 432 and contrasted our study by highlighting the inclusion of second trimester fetal ultrasound data. We have now further enhanced this distinction in the manuscript.

Comment 9: Additionally, limitations include the focus on weight and head circumference at birth without further analysis of other measurements like abdominal circumference, which could strengthen conclusions about FGR patterns in RA and JIA pregnancies.

Response 9: Thank you for this valuable suggestion. While we acknowledge that including abdominal circumference could further enhance our understanding of fetal growth restriction patterns, we focused on weight as a more valid measurement than abdominal circumference at birth. In addition, abdominal circumferences are already included in the computation of Estimated Fetal Weights via the Hadlock formula during second trimester in our study. Future studies among infants should ideally employ more accurate measure of actual body composition, i.e., Air Displacement Plethysmography.

References:

  1. Peterson LS, Mason T, Nelson AM, O'Fallon WM, Gabriel SE. Juvenile rheumatoid arthritis in Rochester, Minnesota 1960-1993. Is the epidemiology changing? Arthritis Rheum. 1996 Aug;39(8):1385-90. doi: 10.1002/art.1780390817. PMID: 8702448.
  2. Crowson CS, Matteson EL, Myasoedova E, Michet CJ, Ernste FC, Warrington KJ, Davis JM 3rd, Hunder GG, Therneau TM, Gabriel SE. The lifetime risk of adult-onset rheumatoid arthritis and other inflammatory autoimmune rheumatic diseases. Arthritis Rheum. 2011 Mar;63(3):633-9. doi: 10.1002/art.30155. PMID: 21360492; PMCID: PMC3078757.
  3. Jan P. Vandenbroucke, Erik von Elm, Douglas G. Altman, et al; for the STROBE initiative. Strengthening the Reporting of Observational Studies in Epidemiology (STROBE): Explanation and Elaboration. Ann Intern Med.2007;147:W-163-W-194. [Epub 16 October 2007]. doi:10.7326/0003-4819-147-8-200710160-00010-w1

Reviewer 2 Report

Comments and Suggestions for Authors

The paper describes the effect of RA/JIA and the effect on foetal growth at 18–22-week scan and compared the neonatal birth weight of affected neonates to those of low risk populations. In addition, they compared a subgroup of patients using medications for RA/JIA and its effect on EFW and neonatal birth weight. 

The study is well designed, and easy to read and understand but to my opinion it doesn't add enough of new knowledge about te effects of autoimmune diseases on foetal growth.

My problem with this study is that they used 18-22 weeks scan as a reference point for evaluating the effect on foetal growth. A later ultrasound should have been used like 28 to 32 weeks since it is very rare to diagnose early foetal growth retardation at 18 – 22 weeks, if it is diagnosed it is usually with chromosomal, syndromes or early onset preeclampsia. But most probably in the health care system that the author’s are from the 18- 22 weeks scan is obligatory, and the data was readily available through their national database. 

Since the patients with autoimmune diseases are known to be high risk patients in pregnancy for several reasons it is unclear to me why data from third trimester was not used or even better why serial ultrasound monitoring was not reported. It would have given us more information about the effect of disease or the use of medication on EFW and neonatal birth weight. The authors address this issue as well line 319-322.

I recommend to the author’s if they wish to keep the paper with the existing data to try publishing in a journal of a lower IP or add additional data from third trimester ultrasound scans.

Author Response

Comment: The paper describes the effect of RA/JIA and the effect on foetal growth at 18–22-week scan and compared the neonatal birth weight of affected neonates to those of low risk populations. In addition, they compared a subgroup of patients using medications for RA/JIA and its effect on EFW and neonatal birth weight. The study is well designed, and easy to read and understand but to my opinion it doesn't add enough of new knowledge about the effects of autoimmune diseases on foetal growth. My problem with this study is that they used 18-22 weeks scan as a reference point for evaluating the effect on foetal growth. A later ultrasound should have been used like 28 to 32 weeks since it is very rare to diagnose early foetal growth retardation at 18 – 22 weeks, if it is diagnosed it is usually with chromosomal, syndromes or early onset preeclampsia. But most probably in the health care system that the author’s are from the 18- 22 weeks scan is obligatory, and the data was readily available through their national database. Since the patients with autoimmune diseases are known to be high risk patients in pregnancy for several reasons it is unclear to me why data from third trimester was not used or even better why serial ultrasound monitoring was not reported. It would have given us more information about the effect of disease or the use of medication on EFW and neonatal birth weight. The authors address this issue as well line 319-322.

Response: Thank you for the feedback. As mentioned in our response to Reviewer 1 above, we agree that including additional fetal ultrasound biometrics data may enhance the understanding of fetal growth trajectories in this population. The Danish Fetal Medicine Database contains >90% of all second trimester fetal ultrasound data from all regions in Denmark, as part of the antenatal care.1 In contrast, we expect missing and incomplete data in the first and third trimester fetal ultrasounds, creating complexity in modeling trajectories. In response to Reviewer 1, we have discussed the complexity and possible bias of analyzing first trimester fetal ultrasound data in Denmark for studying week-specific fetal growth measurement. In terms of third trimester fetal ultrasound data, the completeness of data and timing of scans during pregnancy vary in Denmark depending on the individual’s need. Hence, our project has a priori chosen to focus on analyzing the second trimester data (18-22 weeks), which we believe accurately reflects the fetal growth parameter at this gestational period during pregnancy. Note that a prior Danish study that Reviewer 1 mentioned (Rom et al. 2014), as well as several papers of fetal growth conducted in other populations only analyzed fetal growth parameters measured at birth. Due to funding and resource limitations, at the present stage, our protocol precluded us from extracting the first and third trimester fetal ultrasound data for analyses. Despite this, we contend that our study is the first to detail second trimester fetal ultrasound biometric measurements among offspring of women with RA and JIA using a nationwide study approach. Even without direct measurement of the third trimester data, our results provide important scientific value to advance to field by providing novel results of second trimester fetal ultrasound data and pinpointing that the third trimester development is an important period to be investigated in the future. Third trimester fetal ultrasounds are not routinely performed among patients with RA and JIA in most other nations, including the U.S. To reflect the reviewer’s concerns, we have added the importance in future study to further incorporate fetal biometric measurements from other trimesters to understand fetal growth trajectories (in page 11, lines 438-446 and page 12, lines 478-480). Our research team is seeking further support and resources to conduct further research on this topic. We firmly believe that our results would advance the field and inspire future projects by incorporating additional fetal ultrasound biometric measurements when investigating fetal growth in this population.

Reference:

1. C.K. Ekelund, T.I. Kopp, A. Tabor and O.B. Petersen, '"The Danish Fetal Medicine database," Clin.Epidemiol., vol. 8, Oct 25, pp. 479–483.

Round 2

Reviewer 2 Report

Comments and Suggestions for Authors

I hope that you will continue this study further the evidence you have presented was in a large part expected but has not been published ( to my best knowledge) in this manner. Future research has to be focused on third trimester as stated in your discussion and conclusion. Congratulations on your good work. Regards